# Assessment of the Agricultural Effectiveness of Biodegradable Mulch Film in Onion Cultivation

**DOI:** 10.3390/plants14152286

**Published:** 2025-07-24

**Authors:** Hyun Hwa Park, Young Ok Kim, Yong In Kuk

**Affiliations:** Department of Bio-Oriental Medicine Resources, Sunchon National University, Suncheon 57922, Republic of Korea; camellia9720@nate.com (H.H.P.); 96kimyy@hanmail.net (Y.O.K.)

**Keywords:** biodegradable film, chemical property, onion, polyethylene film, physical property

## Abstract

This study conducted a comprehensive evaluation of the effects of biodegradable (BD) mulching film in onion cultivation, with a focus on plant growth, yield, soil environment, weed suppression, and film degradation, in comparison to conventional polyethylene (PE) film and non-mulching (NM) treatment across multiple regions and years (2023–2024). The BD and PE films demonstrated similar impacts on onion growth, bulb size, yield, and weed suppression, significantly outperforming NM, with yield increases of over 13%. There were no consistent differences in soil pH, electrical conductivity, and physical properties in crops that used either BD or PE film. Soil temperature and moisture were also comparable regardless of which film type was used, confirming BD’s viability as an alternative to PE. However, areas that used BD film had soils which exhibited reduced microbial populations, particularly Bacillus and actinomycetes which was likely caused by degradation by-products. BD film degradation was evident from 150 days post-transplantation, with near-complete decomposition at 60 days post-burial, whereas PE remained largely intact (≈98%) during the same period. These results confirm that BD film can match the agronomic performance of PE while offering the advantage of environmentally friendly degradation. Further research should optimize BD film durability and assess its cost-effectiveness for large-scale sustainable agriculture.

## 1. Introduction

Cultivation which uses mulching induces microclimatic alterations in soil, crops, and the surrounding air environment, ultimately influencing crop productivity [1]. This practice results in various environmental modifications, including the regulation of air and soil temperature, alteration in solar radiation absorption, reduction in water evaporation rates, and control of gas exchange between soil and air. These characteristics can enhance the growth environment of crops, thereby improving productivity. Traditionally, organic mulching materials such as rice straw, wheat straw, wood chips, and sawdust have been employed; however, in recent years, synthetic mulching materials have become predominant. Notably, polyethylene (PE) plastic film is a widely used synthetic mulching material and has become a common agricultural practice in numerous countries, including Korea and China [2,3]. The use of PE film in mulching offers benefits such as suppression of moisture evaporation, warming of soil temperature, weed suppression, reduction in harvest time, prevention of inorganic nutrient leaching, and increased crop productivity [4,5,6]. It is particularly advantageous in regions with limited irrigation or low temperatures [3]. However, PE film is not biodegradable and requires direct recovery post-use [7]. If left on the farmland, PE film takes over 100 years to decompose completely, potentially exerting long-term adverse effects on soil restoration and agricultural ecosystems [8,9,10]. Additionally, incineration of PE mulching film may release dioxins, furans, and other harmful substances, causing environmental pollution and damaging human health [8,10,11,12]. Consequently, biodegradable plastic film is gaining attention as an alternative that addresses environmental pollution concerns while retaining the advantages of PE film [13]. BD plastic film decomposes into carbon dioxide, methane, water, and other substances through microbial action, eliminating the need for separate recovery and disposal processes [14,15,16]. Currently, the predominant BD plastic materials in use are PBAT (poly-butylene adipate terephthalate) and PLA (polylactic acid), each constituting 19% of the BD plastic market [17,18]. PLA is characterized by high tensile strength and processability; however, it is susceptible to heat and prone to tearing, necessitating its combination with PBAT, which offers superior flexibility [19,20]. Furthermore, the development of films utilizing BD materials such as starch blends, PHA, and PHB is being actively pursued in the United States, Japan, and Europe [21,22,23,24,25]. Research into applying these materials to the cultivation of various crops, including tomatoes, corn, garlic, and potatoes [7,26,27,28,29,30], is ongoing. In Korea, investigations are underway to assess the efficacy of BD films for crops such as garlic, deodeok (*Codonopsis lanceolate* (Siebold & Zucc.) Benth. & Hook.f. ex Trautv.), beans, sweet potatoes, and corn [7,31,32,33,34].

In the context of onion (*Allium cepa*) cultivation, photosynthesis is enhanced under long-day conditions, leading to increased nutrient accumulation and promoting the formation and enlargement of bulbs [10]. Moreover, onions are shallow-rooted crops with high sensitivity to moisture stress and weed competition, making topsoil mulching particularly important for ensuring optimal growth. These characteristics, along with the crop’s economic importance and widespread cultivation across temperate regions, make onion a suitable model for evaluating the agronomic and environmental performance of mulching films. Despite growing interest in BD mulching film, studies specifically targeting its use in onion cultivation remain scarce. Furthermore, concerns persist about the rate and by-products of BD film decomposition, including the formation of microplastics and their potential impact on soil health and microbial communities [16,24,25,35,36]. Current research has primarily focused on short-term crop productivity rather than long-term ecological implications, leaving a gap in understanding the environmental trade-offs of replacing conventional PE film with BD alternatives. Therefore, this study applies a recently developed BD mulching film to onion cultivation to evaluate its effects on crop growth and yield, and to analyze the film’s degradation behavior in order to assess its potential environmental impact. By doing so, this research seeks to provide a more holistic understanding of the suitability of BD film for sustainable onion cultivation and its implications for long-term soil quality.

## 2. Materials and Methods

### 2.1. Experimental Site and Mulching Materials

The experiment was conducted in 2023 and 2024 in Seocheon, Chungcheongnam-do (latitude 126.67, longitude 36.14) and Yeongam, Jeollanam-do (latitude 126.68, longitude 34.93). Each test plot was established with a minimum area of 50 m × 20 m per treatment. A total of 600 kg of onion-specific fertilizer (N-P_2_O_5_-K_2_O-MgO-B-Chiyoda: 13-6-8+2+0.2, Chobi Co., Ltd., Seoul, Republic of Korea) per hectare was applied, adhering to the standard application rate [37]. The fertilizer was thoroughly incorporated into the soil prior to mulching. Subsequently, furrows were created using a management machine (Hwangso Agricultural Machinery Co., Ltd., Taegu, Republic of Korea) with a row spacing of 120 cm and planting intervals of 30 cm along each row. Holes were manually made in the mulch at these intervals using tools such as a dibber, and onion seedlings were then transplanted manually into the prepared holes. The experimental treatments included non-mulching (NM), low-density polyethylene (PE) film, and biodegradable (BD) film. All mulching films were black with a thickness of 0.015 mm. The BD film was composed primarily of PBAT and PLA, while the PE film (Seoha P&D Co., Ltd., Damyang, Republic of Korea) was procured from general farm supply stores. In the NM treatment area, manual weeding was performed until March, coinciding with the mid-growth stage of onions, after which the area was left to natural conditions. Following the mulching process, the “Big Boss” and “Yawang” onion varieties were transplanted on 30 October and 13 November, respectively, in the years 2023 and 2024 at the Chungnam test site. Similarly, the “Oreo” and “Katamaru” varieties were transplanted on 8 November and 1 November, respectively, in 2023 and 2024 at the Jeonnam test site. In both locations, seedlings were transplanted approximately 50 days post-sowing, with an inter-plant spacing of 15 cm × 18 cm within the furrows. The chemical composition of the experimental field soil prior to the 2024 trials is detailed in Table 1. Harvest was conducted on 23 May in Yeongam and on 7 June in Seocheon in 2024. The soil type in both experimental sites was sandy loam soil. All other management practices adhered to the guidelines set forth by the Rural Development Administration Standard Management Act [37]. During the experimental periods from October to June of 2022–2023 and 2023–2024, the average air temperatures were very similar between Seocheon (9.04 °C and 9.87 °C) and Yeongam (9.87 °C and 10.99 °C). The average minimum temperatures were also comparable between Seocheon (3.68 °C and 5.12 °C) and Yeongam (3.48 °C and 5.48 °C). However, cumulative precipitation differed notably: in Seocheon, it was 390 mm in 2022–2023 and 267 mm in 2023–2024, while in Yeongam, it was 547 mm and 717 mm, respectively, over the same periods.

### 2.2. Effects of Biodegradable Mulching Film on Onion Growth, Quantity, Weed Occurrence and Post-Crops

In the years 2023 and 2024, the plant height of onions in experimental sites was assessed at 90, 120, 150, 180, and 210 days after transplantation. During the harvest period, several parameters were evaluated, including bolting, bulb height, bulb width, bulb weight, total yield, commercial yield (specifically those exceeding 3 cm in diameter), and commercial yield ratio. Plant height and yield components were measured using 10 randomly selected plants per replication, with three replications for each treatment. However, total yield was measured based on a 3.3 m^2^ area per replication for each treatment. In 2024, we assessed the number of weeds that had grown through tears in the film and weeds that had grown directly through the film. We also assessed weed occurrence (%) in 1000 planting holes that were prepared for onion cultivation, across three replicates. Following the onion harvest in 2024 (210 days post-transplantation), 1 kg of soil was collected from each treatment area to evaluate the impact of biodegradable mulching film on the subsequent crop. The collected soil was placed into a container with a capacity of 500 mL, and a 78.5 cm^2^ section of film was collected from the treatment area and manually cut into approximately 2 mm^2^ pieces using sterilized scissors, which were then thoroughly mixed into the soil. Six soybean seeds were sown immediately after mixing the mulch fragments into the soil in each pot (500 mL). Germination rates were assessed on the seventh day after sowing. Plant height and shoot fresh weight were measured 14 days after sowing.

### 2.3. Degradation Characteristics of Biodegradable Mulching Film During and After Onion Cultivation

In this study, the light transmittance of the film prior to mulching was assessed, and the variation in light transmittance was monitored from 30 days to 210 days at 30-day intervals following onion transplantation. The transmittance was measured using a portable quantum sensor (SKP2200, Skye Instruments, Powys, UK) by collecting three 20 cm^2^-sized pieces of film in triplicate per treatment, cleaning them of dust, and irradiating them with light (200 μmol m^−2^ s^−1^) indoors [38]. Additionally, the degree of degradation of the biodegradable mulching film was evaluated from 30 days post-transplantation to 210 days at 30-day intervals. The degree of film collapse was assessed visually using a scoring scale from 0 to 5. A score of 0 indicates the film remains in its original state; a score of 1 indicates the initiation of film fragmentation; a score of 2 indicates 25% of the film has decomposed into small fragments; a score of 3 indicates the film fragments have decomposed to sizes of 2.0–2.5 cm; a score of 4 indicates the film has deformed into a uniform mesh shape and is no longer in its original state; and a score of 5 indicates the film has decomposed into pieces measuring 4 × 4 cm^2^ or less and is nearly completely degraded [39]. The decomposition rate of the film was evaluated using the film employed for light transmittance irradiation and was calculated based on the difference between the weight of the original mulching film and the weight of the film collected each day after transplanting.

Physical property assessments of the 2024 films were conducted on pre-mulching films and at 150 and 210 days after transplantation. Measurements of tensile strength, tear strength, and elongation were performed utilizing a Texture Analyser (TA-XT plus, Stable Micro System, Godalming, Surrey, UK) as described by Lee et al. [31]. Tensile strength and elongation were evaluated by preparing a film specimen in the Machine Direction (MD) in accordance with ASTM D882 standards, with a strain rate set at 500 mm/min. Tear strength was similarly assessed by preparing a film specimen in the MD direction following KS M 3001 guidelines, maintaining the same strain rate of 500 mm/min. Each physical property measurement was repeated nine times per sample, with the average calculated from seven measurements, excluding the highest and lowest values.

To assess the degree of film decomposition post-onion harvest, three films per treatment were sectioned into 20 × 20 cm pieces at 210 days after transplantation, with measurements made in triplicate. Film degradation experiments were executed under two conditions: one where the film was affixed to the soil surface, and another where the film was buried 10 cm beneath the soil surface. The extent of film decomposition under each treatment condition was evaluated through visual assessment using a scale from 0 to 100, where 100 indicated complete decomposition. Observations were conducted at 30 and 60 days after treatment.

### 2.4. Effects of Biodegradable Mulching on Film Soil Properties

In this study, various soil properties were assessed to determine the impact of biodegradable mulching films on the soil environment. Measurements of soil pH and electrical conductivity (EC) were conducted from day 60 to day 210 at 30-day intervals following transplantation. For this purpose, soil samples were collected from a depth of up to 15 cm from the topsoil of each treatment plot. Specifically, 100 g samples were obtained from three locations per treatment plot, combined, and utilized for analysis. The collected soil was air-dried for five days in a well-ventilated area, sieved through a 2 mm mesh, and subsequently used. Thereafter, 10 g of soil was placed into a 100 mL flask, to which 50 mL of distilled water was added, and the mixture was agitated for 30 min using a reciprocating shaker (Shaking Incubator, HB-201SF, HANBECK SCIENTIFIC TECHNOLOGY, Daejeon, Republic of Korea). Finally, measurements were taken using a pH and EC meter (Portable pH/EC/TDS/Temperature Meter, HI991300, HANNA Instruments, Seoul, Republic of Korea) [38].

The survey of inorganic nutrients in soil was conducted 210 days post-transplantation. Organic content was quantified using the Tyurin method [40]. Specifically, 5 g of soil sample was immersed in a solution containing 0.33 M acetic acid, 0.15 M lactic acid, 0.03 M ammonium fluoride (NH_4_F), 0.05 M ammonium sulfate [(NH_4_)_2_SO_4_], and 0.2 M NaOH (NaOH) in 20 mL, and decomposed for 4 h in a heat storage-type heater maintained at 400 ± 20 °C. Subsequently, to determine the soluble phosphoric acid (P_2_O_5_) content in the extracted solution, analysis was conducted at 470 nm using an ultraviolet (UV) spectrophotometer (UV-1601; Shimadzu, Tokyo, Japan). The concentrations of potassium (K), calcium (Ca), and magnesium (Mg) were extracted by placing 5 g of soil sample into 50 mL of a 1.0 M ammonium acetate (pH 7) solution, followed by measurement using an inductively coupled plasma atomic emission spectrometer (ICP-AES Integra XL; GBC Inc., Arlington Heights, IL, USA) [41]. The content of nitrate nitrogen (NO_3_–N) was leached using a 2 M KCl solution and subsequently analyzed by means of colorimetric methods.

Microbial analysis samples were prepared by collecting soil from a depth of 10 cm below the surface layer at locations devoid of crop roots, using a Combination Soil Auger (ROTAL Eilkelkamp, Giesbeek, The Netherlands). The samples were then sieved through 2 mm mesh. A 10 g portion of the collected soil was added to 90 mL of sterile distilled water and agitated for 10 min at 25 °C and 220 rpm using a reciprocating shaker (Shaking Incubator, HB-201SF, HANBECK SCIENTIFIC TECHNOLOGY, Daejeon, Republic of Korea). The process of adding 1 mL of the shaken dilution to 9 mL of sterile distilled water was repeated to achieve a dilution suitable for microbial enumeration. Bacterial density was assessed using serial dilution, with 100 μL of soil dilution dispensed onto the medium and spread evenly with a sterilized triangular glass rod until no moisture was visible. The inoculated petri dish was sealed and incubated at 28 °C until microbial colonies developed. The resulting colonies were quantified and expressed as colony-forming units (cfu). Aerobic and Gram-negative bacteria were assessed on day 6 post-inoculation using YG medium, while *Bacillus* sp. bacteria were inoculated using YG medium following heat treatment of the prepared soil dilutions in an 80 °C water bath for 20 min, and assessed on day 2. Actinomycetes were evaluated 7 days after inoculation using starch–casein medium, and filamentous fungi were examined 6 days after inoculation on Rose Bengal medium.

Additionally, soil temperature and moisture content were automatically recorded at 30 min intervals throughout the experimental period using a meteorological meter (HOBO USB MicroStation Data Logger, ONSET, Bourne, MA, USA) installed in each treatment unit.

### 2.5. Experimental Design and Statistical Procedures

A completely randomized design was employed for this study, with three replications per treatment. Each treatment plot had a minimum area of 50 m × 20 m, and within each plot, the entire area was uniformly treated with a single type of mulch—no inter-row variation in mulching was applied. For growth parameters such as plant height and yield components, data were collected from 10 randomly selected plants within each of the three replication plots per treatment, for a total of 30 plants per treatment. These 10 plants were randomly sampled from the larger plot area to ensure representativeness and avoid sampling bias. For yield measurements, the experimental unit consisted of plants grown within a 3.3 m^2^ area, selected from within each replication plot. Three such replications were used for each treatment. Statistical significance was evaluated using analysis of variance (ANOVA), and treatment means were compared using Duncan’s Multiple Range Test (DMRT) at *p* = 0.05 [42]. Additionally, independent samples *t*-tests were applied to specific parameters where appropriate.

## 3. Results and Discussion

### 3.1. Effects of Biodegradable Mulched Film on Onion Growth, Quantity and Post-Crop

In both 2023 and 2024, onion plant height was evaluated every 30 days from 90 to 210 days after transplanting in Seocheon and Yeongam under three treatments: biodegradable (BD) film, polyethylene (PE) film, and non-mulching (NM) (Figure 1). Overall, no significant differences in plant height were detected between PE and BD treatments in most observations across both regions and years. In contrast, crops grown in NM plots consistently exhibited significantly shorter plant height, particularly in the Yeongam region. In Seocheon, significant differences among treatments were observed only at 150 days after planting (DAP) in 2023 and 180 DAP in 2024, where PE or BD films resulted in taller plants than NM. However, these differences disappeared by the harvest period (210 DAP). In Yeongam, similar trends were observed, with NM plots producing shorter plants across most time points, while PE and BD treatments were statistically similar. Notably, at early stages (e.g., 90–120 DAP), plant height varied slightly between PE and BD, but these differences were not consistent over time. These findings suggest that BD film supported onion growth comparably to PE film in both regions, likely due to similar warming and moisture-retaining effects, although statistical equivalence cannot be conclusively inferred.

In this study, we examined the bolting rate, bulb height, bulb width, bulb weight, yield, commercial yield, and commercial yield ratio for each harvest season, categorized by film type (Table 2). The bolting rate and bulb height were not significantly different regardless of whether NM, BD, and PE treatments were used in all survey years and locations. No statistically significant differences were detected between crops treated with PE and BD films in terms of bulb width and bulb weight across all years and locations; however, crops grown without mulch (NM) showed significantly lower values in these components. Similarly, yield differences between PE and BD films were not statistically significant in any year or location. In contrast, NM plots produced significantly lower yields than PE and BD plots in 2024. Commercial yield and commercial yield ratio showed no significant differences regardless of mulching type in all survey periods and locations. These results indicate that while PE and BD films supported comparable trends in yield components, statistical tests did not detect significant differences between them. Meanwhile, crops in NM plots were consistently inferior in several components such as bulb width, bulb weight, and total yield.

According to the weed occurrence survey based on film type, weeds did not emerge at the degraded film areas or through the intact film, irrespective of the region or film type (Table 3). However, in both regions, there was no significant difference in weed occurrence in crops grown using both PE and BD films. In contrast, weeds emerged in 4–5% of the planting holes in the NM areas, which was significantly higher than the 1–2% observed in PE mulch and 0–3% in BD. The predominant weed species included *Capsella bursa-pastoris*, *Portulaca oleracea*, *Acalypha australis*, *Chenopodium album*, *Artemisia princeps*, *Erigeron canadensis*, *Lamium amplexicaule*, and *Humulus japonicus.* Although the differences in weed suppression between PE and BD films were not statistically significant, the slightly higher weed occurrence in BD treatments (0–3%) compared to PE (1–2%) could suggest minor variations in physical durability or film–soil adhesion. However, from a practical standpoint, these small differences are unlikely to require different management practices. In this study, no dominant weed species were observed to consistently penetrate or resist the BD film. The weeds that did emerge were typically located in planting holes or film tear points, indicating that the BD film was effective in suppressing weed emergence across all dominant species identified.

Following onion harvesting, residual film and soil were collected from PE and BD areas, while only soil was collected from the NM areas. Soybeans were subsequently sown in pots containing collected soil as a post-crop to assess initial growth (Figure 2). The survey results indicated that the germination rate of soybeans was lower in soil from areas that used NM compared to soil from areas that used PE and BD films across all regions, with seeds planted in soil from areas that used PE film also exhibiting a lower rate of germination than soybean seeds planted in soil from areas where BD film was used. In Seocheon, the plant height of soybean plants grown in soil from NM areas was significantly taller than those grown in soil from areas that used PE and BD films, with no difference between soybeans grown in soil from PE and BD areas. Conversely, in Yeongam, soybean plant height of plants grown in soil from NM areas was shorter than plants grown in soil from PE and BD areas, with similar trends observed between PE and BD. The aboveground biomass of soybean plants grown in soil collected from Seocheon was significantly greater when plants were grown in soil from NM areas compared to plants grown in soil from PE and BD areas, mirroring the plant height results. However, in Yeongam, no significant differences in aboveground biomass were observed in plants grown in soil from NM, PE, and BD areas. These contrasting results between the Seocheon and Yeongam sites may be attributed to several factors. First, differences in initial soil fertility—as indicated by higher organic matter and available phosphorus in Seocheon (Table 1)—could have influenced soybean early growth responses. Second, regional variations in microbial activity and climate conditions (e.g., temperature and moisture) may have affected the rate and nature of BD film decomposition, potentially altering soil chemical or biological conditions post-harvest. Although no direct negative impact from BD film degradation by-products was observed, site-specific microbial community dynamics might have interacted differently with residual film particles, thereby affecting plant performance. Further analysis of soil microbial composition and enzyme activity would help clarify these site-specific outcomes in future research. Overall, while germination rates tended to be highest in BD-treated soils and lowest in NM soils, plant height and biomass responses to soil type were not consistent across regions, and no significant differences were observed between PE and BD treatments.

In this study, it was determined that the recently developed BD film did not adversely affect the growth, yield, weed occurrence, or post-harvest condition of winter crops such as onions, when compared to commercial PE film. Similarly, when BD film was applied to summer crops, including corn, tomatoes, soybeans, and cotton, no significant differences in growth and yield were observed compared to PE films [7,27,28,29,30,38]. Furthermore, consistent with the findings of this study, when the BD film was incorporated into the soil for the cultivation of soybeans, barley and oats, no adverse effects on post-crop growth were noted [43,44]. Conversely, certain biodegradable plastic extracts were found to reduce germination in lettuce (B-SP4) and tomatoes (B-SP4 and B-SP-6), and significantly hinder root development in lettuce [45]. This study is significant in that it confirms the potential of BD film to contribute to sustainable agriculture. Future economic analyses, alongside further validation across diverse crops and environmental conditions, are anticipated to enhance the feasibility of utilizing BD.

### 3.2. Degradation Properties of Biodegradable Mulch Film During and Following Onion Cultivation

This study evaluated the light transmittance of BD and PE films on onion fields in 2023 and 2024, from 30 to 210 days after transplantation (Figure 3). In Seocheon (2023), no difference was observed early on, but BD film showed significantly higher transmittance at 180 and 210 days. In Yeongam (2023), BD film consistently exhibited 1–2% higher transmittance than PE from 60 to 210 days. In 2024, BD film in both regions showed significantly higher transmittance than PE throughout the measurement period. PE film reached only 0.5% or less, while BD film reached up to 2.5%. These differences likely stem from variations in growing periods and environmental temperatures. In contrast, BD film used in soybean (6.4–15.8%) [38] and lettuce (4.3% to 9.4%) [46] cultivation showed much higher transmittance due to faster degradation under warmer conditions. The relatively low transmittance in onions is likely due to cooler temperatures slowing film decomposition and crack formation. These findings underscore the importance of considering crop season, environmental temperature, and film properties in BD film application.

From 30 to 210 days after transplantation, film degradation on onion fields was visually assessed using a 0–5 scale (Figure 4 and Figure 5). In 2023, BD film degradation began at 150 days in both Seocheon and Yeongam, reaching score 1 (film fragments) by day 210. In 2024, BD film degraded more than PE film from 150 to 210 days, though degradation was less pronounced than in 2023, likely due to weather differences. The slower breakdown is attributed to reduced microbial activity under cooler conditions during onion cultivation. In contrast, faster degradation in summer crops like soybeans is promoted by higher temperatures that enhance microbial activity [38].

Film degradation rates were assessed in Seocheon and Yeongam during 2023 and 2024 (Figure 5). In 2023, no significant difference was observed between BD and PE films overall, but in Yeongam, BD film showed significantly higher degradation from 120 to 210 days after transplantation. At day 210, PE film degradation was <1%, while BD film reached ~8%, based on weight loss. This elevated degradation in Yeongam may reflect local factors such as temperature, soil moisture, and microbial activity [26,28]. Maintaining an appropriate degradation rate is critical—too fast and weed suppression and soil insulation may be reduced, while if too slow residual film may be left, risking environmental harm. For the BD film used here, a slightly higher degradation rate may be desirable to balance functionality and sustainability.

In 2024, the tensile strength, elongation, and internal tearing strength of BD and PE films was examined prior to transplantation and at 150 and 210 days following onion transplantation in the Seocheon and Yeongam regions (Figure 6). Regarding tensile strength, in the Seocheon region, both PE and BD films exhibited a significant reduction at 150 and 210 days after transplantation compared to pre-transplantation levels. Notably, the tensile strength of the BD film was markedly lower than that of the PE film. In the Yeongam area, the tensile strength of the PE film was not significantly different pre-transplantation and 150 and 210 days after transplantation. However, the BD film tensile strength was lower than that of the PE film and showed a decreasing trend at 150 and 210 days after transplantation. Nevertheless, no significant difference was observed between 150 and 210 days after transplantation. In the Seocheon region, the elongation of PE film exhibited a significant increase at 150 and 210 days after transplantation compared to pre-transplantation levels. The elongation of the BD film was 239 to 790 times greater than that of the PE film, a disparity that persisted both before transplantation and at 150 and 210 days after transplantation. In the Yeongam area, the trend in elongation rate mirrored that of Seocheon, although the elongation rate of the PE film was marginally higher than in Seocheon. Similarly, the elongation rate of the BD film in Yeongam was 239 to 403 times greater than that of the PE film, consistent with observations in Seocheon. The pronounced elongation of the BD film is likely attributable to the molecular structure characteristics of the film or the inherent properties of BD materials. In the Seocheon region, the tear strength of PE film did not exhibit significant variation before transplantation and at 150 and 210 days after transplantation. Conversely, BD film demonstrated a significant reduction in tear strength at 150 and 210 days after transplantation compared to the pre-transplantation period, although no significant difference was observed between the 150 and 210-day marks. In the Yeongam area, the pattern observed in the tear strength mirrored that of Seocheon. The PE film showed no significant differences before transplantation and at 150 and 210 days after transplantation. However, the BD film exhibited lower tear strength than the PE film prior to transplantation and experienced a significant decrease at both 150 and 210 days after transplantation, with no significant difference between these two time points. The reduction in tear strength of the BD film is attributed to the decline in physical strength as the film undergoes natural decomposition. Collectively, these findings indicate that BD film experiences a rapid decline in mechanical strength, particularly exhibiting a significantly higher elongation rate compared to PE film. This phenomenon is consistent with previous studies documenting the natural degradation of BD film under soil conditions [47,48]. Sintim et al. [47] demonstrated that the tensile strength of polylactic acid (PLA)-based BD mulch film decreased by approximately 40% after six months in field conditions, supporting the rapid weakening of BD film observed in this study. The consistently higher elongation observed in BD film relative to PE film likely reflects the molecular structure and polymer chain flexibility characteristic of biodegradable materials such as polybutylene adipate terephthalate (PBAT) [49]. These results align with those of Ludwiczak et al. [50], who reported that PBAT-based film exhibited elongation rates several hundred times greater than those of conventional PE film, contributing to its distinct mechanical behavior. The observed differences between Seocheon and Yeongam appear to be attributable to environmental factors, including weather conditions, soil characteristics, and microbial activity. BD film is more suitable for rapid biodegradation compared to traditional PE film, particularly in contexts such as summer crop cultivation. However, BD film may present limitations for crops where durability is a critical factor. Consequently, it is imperative to select a film by thoroughly considering the characteristics of the crops, climate conditions, and cultivation period. These findings suggest that the BD film maintained sufficient physical integrity during the early to mid-stages of onion growth, as supported by the low degradation scores and transmittance values prior to 150 days after transplantation. However, in later stages (post-150 days), mechanical properties such as tensile strength and tear resistance declined, and degradation rates increased. While weed suppression and moisture conservation were not negatively affected in this study, there is potential for early degradation to compromise mulching benefits in longer-season crops or under higher microbial activity conditions. Thus, for winter crops like onion, BD film appears adequately durable, but its performance under faster-degrading conditions—such as in warmer summer crops—should be carefully monitored. In warmer environments, higher microbial activity and increased soil temperatures may accelerate film degradation, potentially shortening the duration of mulch effectiveness. Future studies are needed to fine-tune the balance between biodegradation timing and mulching performance, especially in diverse cropping systems and seasonal climates.

During the onion harvest season, BD and PE films were placed at a depth of 10 cm below the soil surface and at the soil surface, respectively, to assess their degradation rates at intervals of 30 and 60 days (Figure 7). In the Seocheon area, the degradation rate of PE film was consistently low, remaining below 3% at both 30 and 60 days after it was applied, irrespective of its placement on the soil surface or at a depth of 10 cm. Conversely, BD film exhibited degradation rates of 24% and 78% at the soil surface and 10 cm depth, respectively, at 30 days after its application, with an increase to 38% and 99%, respectively, after 60 days. Similarly, in the Yeongam area, the degradation rate of PE film remained low, below 3%, regardless of how long it had been applied or the depth of burial. However, BD film demonstrated a degradation rate of 25% or less when buried at a 10 cm depth at 30 days after its application. After 60 days, the degradation rate was 8% or less at the soil surface, while it reached 98% at a 10 cm depth. These findings suggest that BD film exhibits a low degradation rate at the soil surface but undergoes rapid degradation when buried at a 10 cm depth, likely due to active microbial activity in the soil.

### 3.3. Effects of Biodegradable Mulching Film on the Soil Environment

The investigation into soil pH variations on onion plantations utilizing BD and PE mulching yielded the following results (Figure 8). In Seocheon in 2023, 60 days post-transplantation, soil pH in areas where BD film was used was higher than in areas where PE film was used. However, at 90 and 150 days post-transplantation, the soil pH of areas using BD film was lower than in areas that used PE film. In the case of NM, the pH was significantly higher than in areas using both BD and PE films at 180 and 210 days after transplantation, during the later stages of crop growth. In Yeongam, soil pH in areas where BD film was used was significantly higher than areas where PE film was used at 90 days post-transplantation, but tended to be lower than after 180 and 210 days. In Seocheon in 2024, soil pH in areas where BD film was used was significantly lower than areas that used PE film at 90 and 180 days after transplantation, although no significant differences were observed at other time points. For NM, a significantly lower pH was recorded at 90, 180, and 210 days after transplantation compared to areas that used both BD and PE films. In Yeongam, soil pH in areas where BD film was used was significantly higher than in areas that used PE film at 90 and 120 days after transplantation, with no differences observed during other irradiation periods based on mulching type. Furthermore, an examination of the pH of the PE and BD films themselves revealed that the pH of BD film was significantly higher than that of PE film both before transplantation and 120 and 150 days after transplantation. However, despite the high pH of the BD film itself, it does not appear to consistently influence soil pH, suggesting that the chemical properties of the film itself do not significantly affect soil pH. The variations in soil pH between areas where BD and PE films were used differed by time and region, potentially exerting a complex influence on soil chemistry as organic matter degraded over time due to the biodegradation properties of BD film. The tendency for plots without mulch (NM) to exhibit relatively high pH stresses the importance of carefully considering the effects of mulching films on the soil’s redox state and the rate of organic matter degradation. Overall, the difference in soil pH between areas that used BD and PE films was not significant, and the influence of the film’s pH on soil pH was limited.

In examining the soil electrical conductivity (EC) results from the Seocheon area in 2023, it was observed that soils covered with biodegradable (BD) film mulch exhibited lower EC values at 60, 120, 150, and 180 days after transplantation compared to soils mulched with polyethylene (PE) film (Figure 9). However, at 90 days after transplantation, the soil EC under BD film was slightly higher than under PE film, and at 210 days, no difference was observed between the two mulch types. The EC values for non-mulched (NM) soil did not display a consistent trend overall, although at 120 and 150 days post-transplantation, the values were lower than those for PE film. In the Yeongam area, the EC for BD film was lower than that for PE film at 60, 90, and 120 days post-transplantation, but it was higher at 150 days, with no difference observed at 180 and 210 days. The EC for NM soil was consistently lower than the EC for both BD and PE films from 90 to 210 days post-transplantation. In the Seocheon area in 2024, the EC for BD film was higher than that for PE film at 90 and 210 days after transplantation, while no significant differences were observed during other survey periods. The EC for NM soil was lower than BD and PE at 90, 180, and 210 days after transplantation. In the Yeongam area, EC for BD was only significantly higher at 180 and 210 days after transplantation, and there was no significant difference in the remaining survey period. In addition, the EC of NM was not consistently different from the EC of PE and BD. On the other hand, the EC of the film itself was 3–4 times higher in BD than PE before transplantation and at 120 and 150 days after transplantation, but there was no difference in EC between the BD and PE of the soil during that period, so it was determined that that the EC of the film itself did not affect the soil. Taking this into consideration, it appears that the soil environment (moisture, temperature, organic matter, etc.) had a greater impact on EC changes than the physical properties of the film.

In the Seocheon area in 2023, the EC value remained below 1 dS/cm at all survey intervals, except for 180 days after transplantation, suggesting minimal impact on crop growth. A comparable study measured pH and EC in onion plantations utilizing biodegradable mulching film, finding no significant differences between BD and PE films [51]. Similarly, on pumpkin plantations, the use of BD mulching film did not result in significant differences in soil pH and EC compared to PE film [52]. Furthermore, a study involving barley cultivation with ground film mixed into the soil reported no significant differences in soil pH and EC between biodegradable and PE films [44]. Collectively, these studies and the present one indicate that BD film does not significantly affect soil pH and EC, likely due to the biodegradation properties of these films influencing soil chemistry.

The analysis of soil inorganic nutrients from BD and PE treatment plots during the 2024 onion harvest season (210 days after transplant) yielded the following results (Figure 10). In Seocheon, no significant difference was observed in the organic matter content based on the type of film used. Conversely, in the Yeongam region, the organic matter content was significantly lower when BD film was used compared to PE film, and significantly lower in areas with NM compared to both areas using BD and PE films. The nitrate nitrogen content was considerably lower in areas that used BD film than areas that used PE film in Seocheon, with no observed difference between areas that used NM and PE films. In the Yeongam area, no significant difference was found between areas that used BD and PE films, while NM areas exhibited significantly lower levels than areas that used PE and BD films. The available phosphate content followed the order of NM > PE > BD in the Seocheon region, with no significant difference between NM and PE in the Yeongam region, although BD was significantly lower than PE. The levels of calcium and magnesium among the exchangeable cations did not significantly vary with film type in the Seocheon region. In the Yeongam area, no significant difference was noted between PE and BD, but NM was substantially lower than both PE and BD. Regarding potassium content, no significant difference was found between areas that used PE and BD films in the Seocheon region, although it was significantly lower than NM areas. In the Yeongam area, no difference was observed between PE and BD, while NM was significantly lower than both PE and BD.

In general, the nutrient levels in the soil were higher in the Yeongam region compared to the Seocheon region. This discrepancy is attributed to several factors, including initial soil fertility, physical properties, and variations in rainfall and temperature. However, most inorganic nutrients did not exhibit a consistent pattern based on the type of film used. Notably, the effective phosphoric acid content was lower in BD film than in PE film in both regions. In Yeongam, the organic matter content was lower in BD film, and in Seocheon, BD film had lower nitrate nitrogen content compared to PE film. On potato plantations, the organic content of soil using biodegradable mulching film (14.70 g/kg) was significantly higher than that of soil using PE film (12.85 g/kg) [27]. Conversely, on zucchini plantations where BD mulching film was used continuously for four years, the soil organic matter content did not differ significantly from that of PE film [52]. Similarly, on onion plantations, no significant difference in soil organic matter content was observed between BD and PE films [51]. In a related study, garlic and corn plantations using biodegradable mulching film exhibited significantly higher effective phosphoric acid content (25–40 mg/kg) than those using PE film (20–25 mg/kg) [53]. Additionally, on zucchini plantations, no significant difference was found in the content of exchangeable cations such as Ca and Mg between BD and PE films [52]. Thus, across numerous studies including the present one, inorganic nutrient levels did not demonstrate a consistent trend under the conditions of BD and PE films. This inconsistency may be attributed to the differential impact of rainfall and temperature variations in the Seocheon and Yeongam regions, as well as the initial fertility and physical properties of the soil. Furthermore, although this study focused on onions, previous research has reported contradictory results in various crops such as potatoes, pumpkins, and corn [52,53]. These discrepancies may arise from differences in root structure and nutrient uptake characteristics of the crops.

To examine the impact of BD mulching film on the physical properties of soil on onion plantations, parameters such as bulk density, porosity, and the three-phase system were assessed (Figure 11). The investigation revealed no significant differences in soil bulk density based on film type in both the Seocheon and Yeongam regions. Porosity exhibited a notable decrease in areas that used BD film compared to PE film and NM areas. Among the three phases, no significant differences were observed in Seocheon based on film type; however, in Yeongam, BD was significantly reduced compared to PE and NM. Regarding the liquid phase, no differences were noted between film types in Seocheon, whereas in Yeongam, PE showed a significant decrease compared to BD and NM. In Seocheon, BD was less than PE, with no significant difference between BD and PE in Yeongam. Overall, porosity demonstrated a significant decrease in BD compared to PE in both regions, but no consistent pattern emerged in the other factors. Consistent with this study, previous reports have indicated no differences between BD film and PE film in soil, floor area density, porosity, and three phases in soybean, garlic, and corn plantations [38,53]. Conversely, some studies have reported that BD mulching film enhances soil physics by promoting degradation and increasing microbial populations [10,27]. In conclusion, the BD film utilized in this study does not appear to significantly affect soil physics.

In 2024, a survey of soil temperature and moisture content in areas using BD and PE films on onion plantations indicated no differences in soil temperature between BD and PE in both Seocheon and Yeongam areas (Figure 12). While plots without mulch (NM) generally showed lower values, no significant differences were observed between BD and PE, although some survey days recorded slightly lower values. During the onion cultivation period from January to February in the Seocheon region, soil moisture levels were slightly higher in areas that used BD film compared to areas that used PE film. However, from March until the harvest season, areas using PE film generally exhibited higher moisture content than areas using BD film. Throughout the trial period, NM treatment plots consistently demonstrated lower moisture content than both BD and PE. Conversely, in the Yeongam area, no significant difference in soil moisture content was observed between BD and PE, although both treatments exhibited higher moisture content than NM. In studies involving BD film on soybean plantations, no significant difference in soil temperature was found between BD mulching film and PE film. However, mulching treatment was reported to increase soil temperature by approximately 2 °C compared to non-mulching [38]. This effect is likely due to the similar surface coverage provided by BD and PE films, which regulate insolation and prevent geothermal loss. Additionally, soil moisture content increased by approximately 5 to 15% with mulching treatment compared to non-mulching, although the difference between BD and PE films was minimal. These findings align with previous studies indicating that BD and PE exhibit comparable temperature and moisture control effects in onion plantations, which are winter crops [38], suggesting the potential for BD to serve as a substitute for PE. On corn plantations, BD mulching film treatment generally resulted in higher soil temperature and moisture content compared to non-mulching, with no significant difference between BD and PE [54]. However, regional environmental variations may influence the water retention effects of BD and PE, necessitating the selection of appropriate mulching materials based on crop characteristics and cultivation conditions.

In the examination of microbial populations within the BD, PE, and non-mulching (NM) plots of onion plantations, it was observed that aerobic bacteria did not exhibit significant differences by film type in the Seocheon region. However, in Yeongam, the BD treatment demonstrated a lower presence of aerobic bacteria compared to NM and PE, as indicated in Table 4. Bacillus bacteria were significantly less prevalent in BD treatments than in NM and PE in both Seocheon and Yeongam. In Seocheon, actinomycetes were least abundant in BD, followed by NM and PE, whereas in Yeongam, actinomycetes were significantly less in BD and NM compared to PE. Gram-negative bacteria were significantly less abundant in PE and BD than in NM in Seocheon, and significantly less abundant in BD than in NM and PE in Yeongam. Filamentous fungi did not vary according to film type in Seocheon, but in Yeongam, their abundance followed the order of PE > NM > BD. Overall, the microorganisms examined in this study tended to be less abundant in the BD treatment. These reductions may be linked to the physical and chemical changes in the soil caused by the degradation of BD film. The by-products of BD film decomposition, such as organic acids or oligomers from PBAT or PLA, may temporarily alter soil pH or nutrient availability, leading to reduced microbial activity or shifts in community structure [51,52]. The lower abundance of Bacillus and actinomycetes, in particular, is noteworthy, as these microbial groups play crucial roles in organic matter decomposition, nutrient mineralization, and biocontrol of soilborne pathogens [53,54]. A decline in their populations could potentially influence nutrient cycling efficiency, plant health, and soil resilience, especially with repeated use of BD film over multiple growing seasons. Although this study did not detect immediate agronomic consequences such as reduced yield or crop performance, the microbial shifts observed may signal longer-term ecological changes in the soil environment. As such, repeated application of BD film could lead to cumulative effects on microbial-driven soil functions. Previous studies have also shown that BD plastics can alter microbial community composition, sometimes favoring microbial taxa capable of utilizing plastic degradation intermediates, while suppressing others [55,56]. Given these findings, it is essential to investigate the long-term effects of BD film degradation on microbial diversity, functional groups, and soil biochemical processes. Future studies should include metagenomic or enzymatic analyses to characterize microbial community function in more detail and to determine whether BD film has transient or persistent impacts on soil health.

## 4. Conclusions

This study demonstrated that biodegradable (BD) mulching film performs comparably to conventional polyethylene (PE) film in supporting onion growth, yield, and weed suppression across different regions and years. BD and PE films exhibited similar effects on soil pH, electrical conductivity (EC), inorganic nutrients, and physical properties, with minimal differences in temperature and moisture regulation. These findings suggest that BD film can serve as an effective alternative to PE film while promoting sustainable agricultural practices by reducing plastic waste. The degradation of BD film initiated around 150 days post-transplantation, with fragmentation increasing over time. BD film exhibited higher light transmittance and decomposed more rapidly than PE film, particularly when buried at a 10 cm depth, where microbial activity facilitated its breakdown. Mechanical strength assessments indicated that BD film weakened progressively, highlighting the need for optimization to ensure sufficient durability throughout the crop cycle. Despite its accelerated degradation, BD film did not negatively impact post-crop growth, reinforcing its viability for agricultural use. A key finding was the significant reduction in microbial populations—such as Bacillus, actinomycetes, and filamentous fungi—in BD-treated soils compared to PE and non-mulching (NM) plots. This suggests that by-products generated during BD film degradation may influence microbial communities, potentially affecting organic matter decomposition and soil health. However, several limitations should be acknowledged. The study was conducted at only two sites and over two growing seasons, which may limit the generalizability of the findings across broader environmental conditions. Additionally, the degradation scoring system used was based on visual assessment, which is inherently subjective and may benefit from more quantitative or molecular evaluation methods in future studies. Future research should focus on long-term studies evaluating the cumulative effects of repeated BD film use on soil microbial dynamics, nutrient cycling, and crop productivity. Further investigation into the chemical composition of degradation by-products and their ecological impact is also needed. Moreover, economic analyses—such as cost-benefit comparisons between BD and PE films—would provide critical insights for large-scale agricultural adoption. Overall, BD film provides equivalent agronomic benefits to PE film while offering an environmentally sustainable solution to agricultural plastic waste. Yet, balancing biodegradability with functional durability remains essential. Continued refinement of BD film formulations and evaluation under diverse cropping systems will be key to supporting its broader use in sustainable agriculture.

## Figures and Tables

**Figure 1 plants-14-02286-f001:**
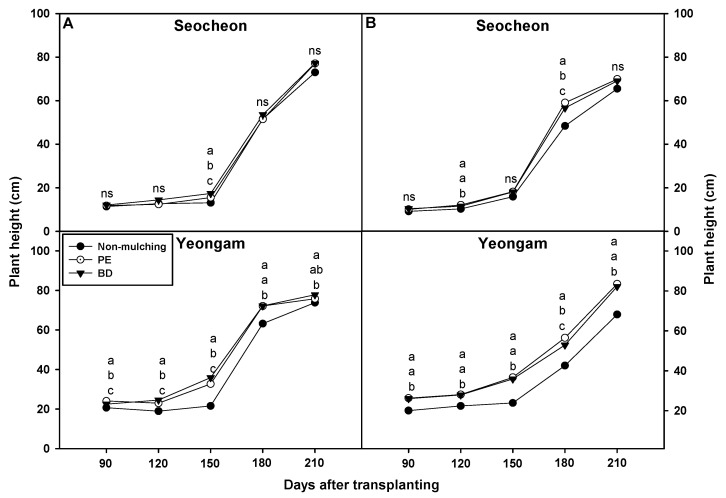
Effect of biodegradable (BD) and polyethylene (PE) films on plant height in onion cultivation areas in 2023 (**A**) and 2024 (**B**). Within each box (representing an area), mean values with the same superscript letter(s) are not significantly different at the 5% level according to Duncan’s Multiple Range Test (DMRT). ns, non-significant.

**Figure 2 plants-14-02286-f002:**
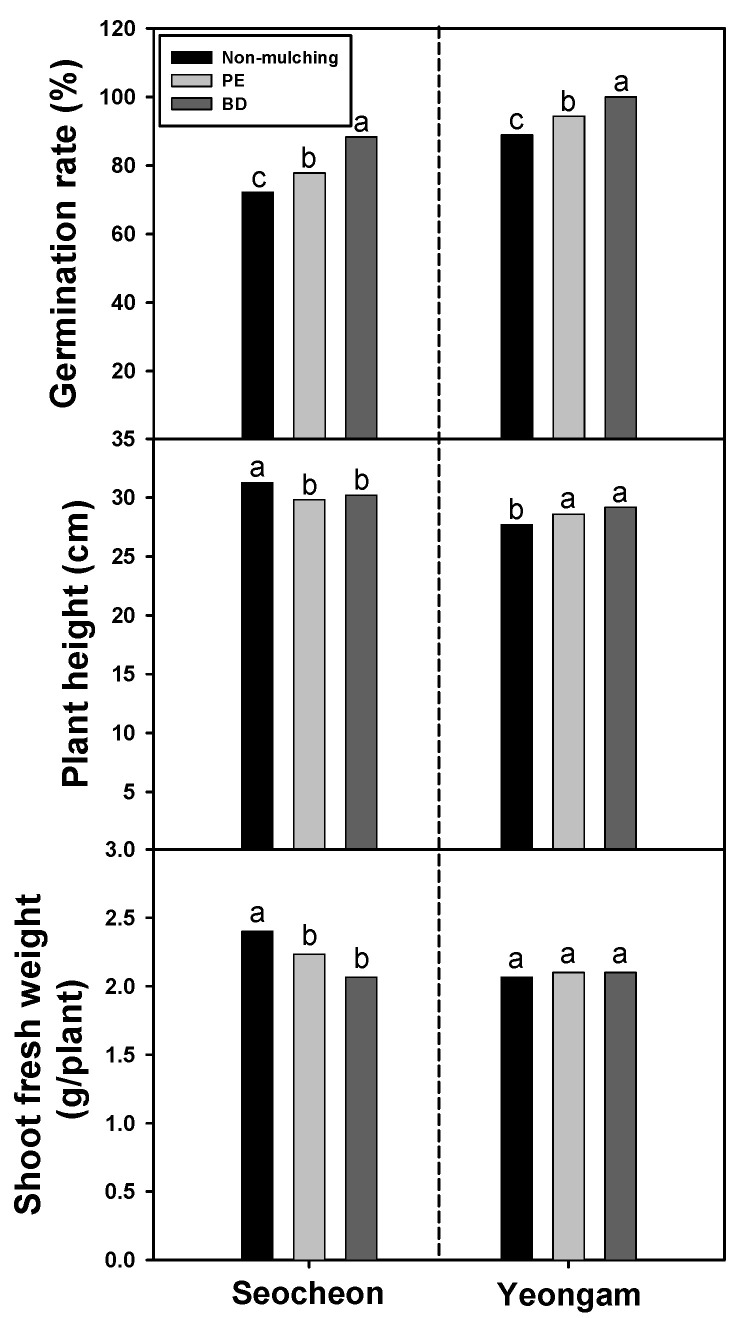
Effect of biodegradable (BD) and polyethylene (PE) films on germination, plant height, and shoot fresh weight of post-crop (soybean) planted in soil collected from onion cultivation areas in 2024. Within each box (representing an area), mean values with the same superscript letter(s) are not significantly different at the 5% level according to Duncan’s Multiple Range Test (DMRT).

**Figure 3 plants-14-02286-f003:**
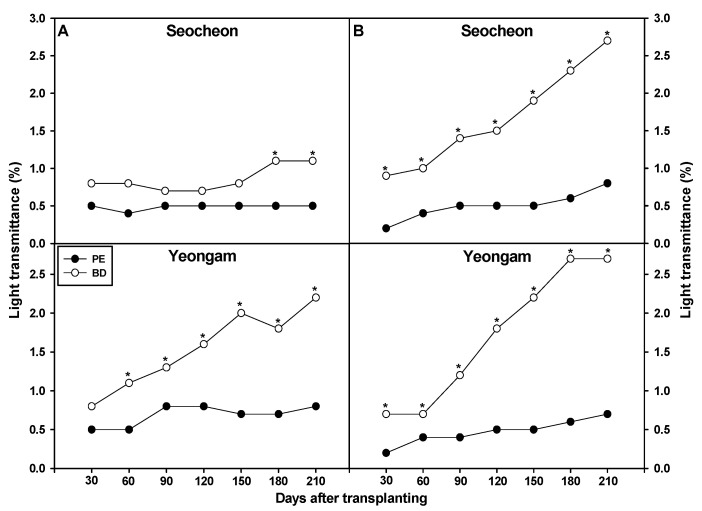
Effect of biodegradable (BD) and polyethylene (PE) films on light transmittance in onion cultivation areas in 2023 (**A**) and 2024 (**B**). * Indicates a significant difference between PE and BF according to a *t*-test (*p* = 0.05).

**Figure 4 plants-14-02286-f004:**
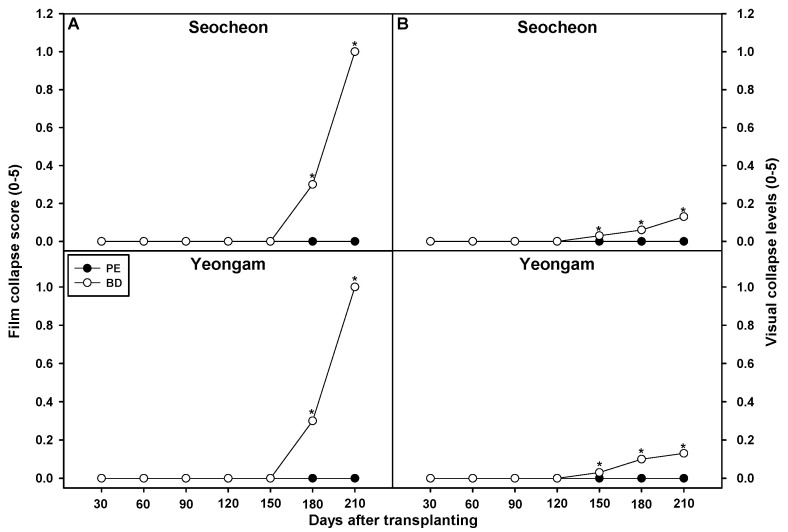
Effect of biodegradable (BD) and polyethylene (PE) films on film collapse scores (0–5) in onion cultivation areas in 2023 (**A**) and 2024 (**B**). * Indicates a significant difference between PE and BF according to a *t*-test (*p* = 0.05). Visual collapse scores (0–5): 0 represents a film that is practically intact, while 5 represents a film that has broken down into fragments smaller than 4 × 4 cm^2^.

**Figure 5 plants-14-02286-f005:**
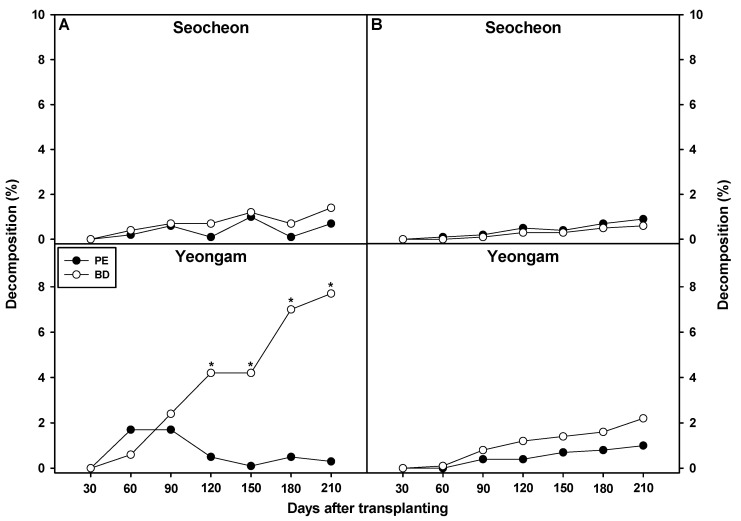
Effect of biodegradable (BD) and polyethylene (PE) films on film decomposition (%) in onion cultivation areas in 2023 (**A**) and 2024 (**B**). * Indicates a significant difference between PE and BF according to a *t*-test (*p* = 0.05).

**Figure 6 plants-14-02286-f006:**
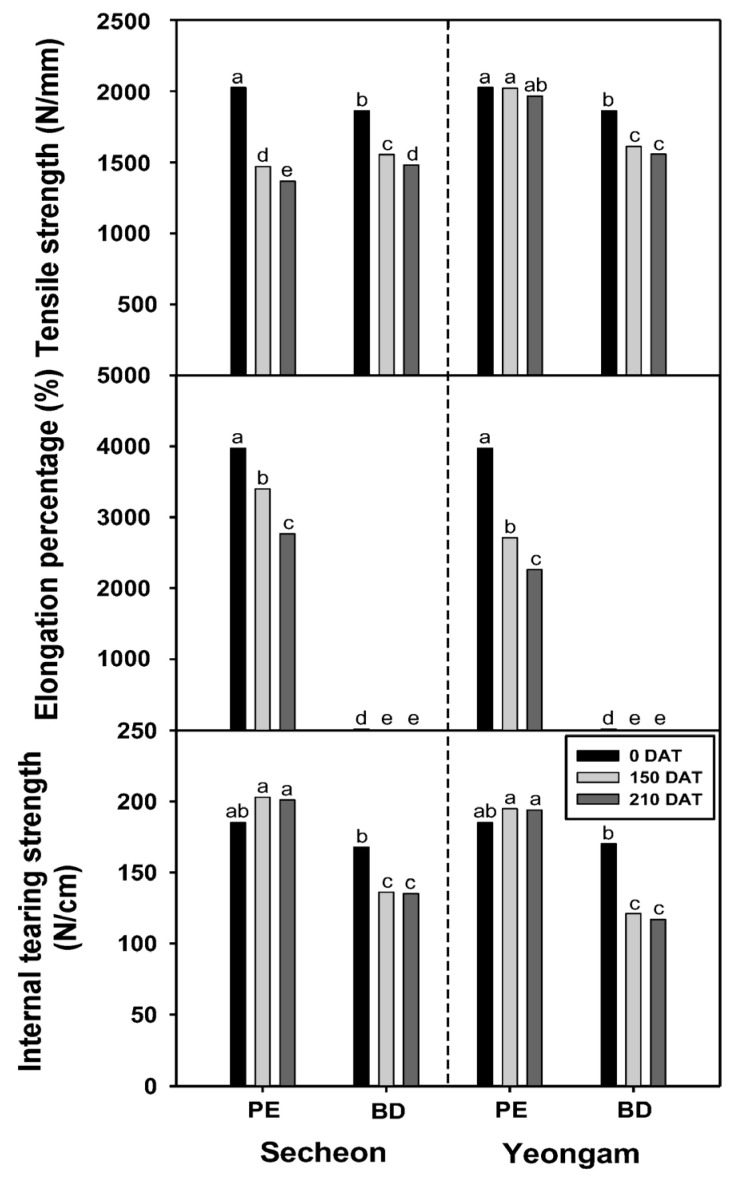
Tensile strength, elongation percentage, and internal tearing strength of biodegradable (BD) and polyethylene (PE) films in onion cultivation areas in 2024. Within each box (representing an area), mean values with the same superscript letter(s) are not significantly different at the 5% level according to Duncan’s Multiple Range Test (DMRT). DAT, days after transplanting.

**Figure 7 plants-14-02286-f007:**
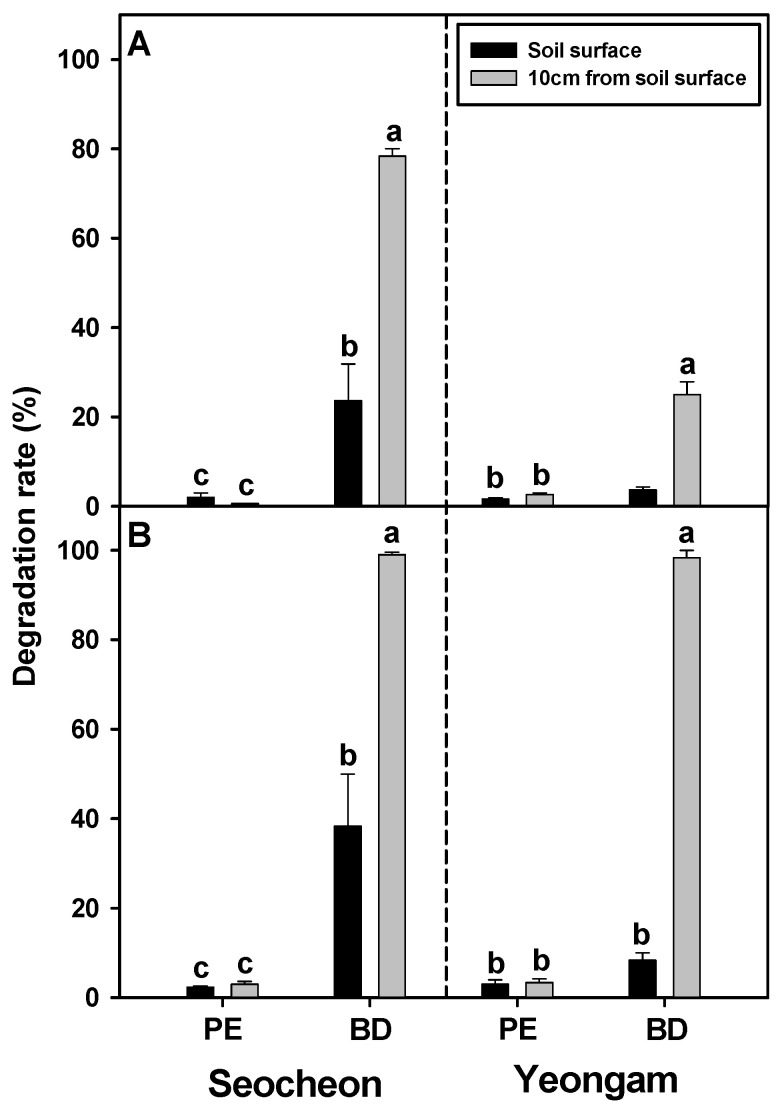
Degradation rate (%) of biodegradable (BD) and polyethylene (PE) films on the soil surface and at a 10 cm soil depth according to the duration after onion harvest ((**A**), 30 DAH; (**B**), 60 DAH) in 2024. Within each box (duration after onion harvest), mean values with the same superscript letter(s) are not significant different at the 5% level according to Duncan’s Multiple Range Test (DMRT). DAH, days after harvesting.

**Figure 8 plants-14-02286-f008:**
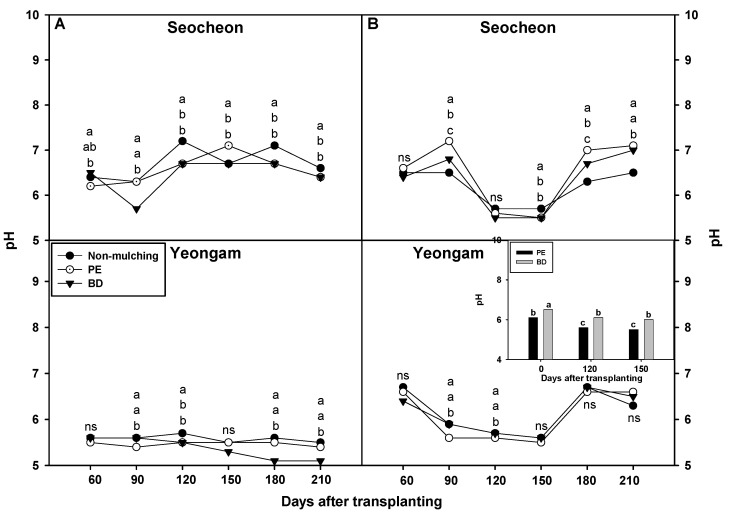
Effect of biodegradable (BD) and polyethylene (PE) films on soil pH in onion cultivation areas in 2023 (**A**) and 2024 (**B**). The values inside the figure represent the pH of the film solution at 120 and 150 days after transplanting in Yeongam, 2024. Within each box (representing an area), mean values with the same superscript letter(s) are not significantly different at the 5% level according to Duncan’s Multiple Range Test (DMRT). ns, non-significant.

**Figure 9 plants-14-02286-f009:**
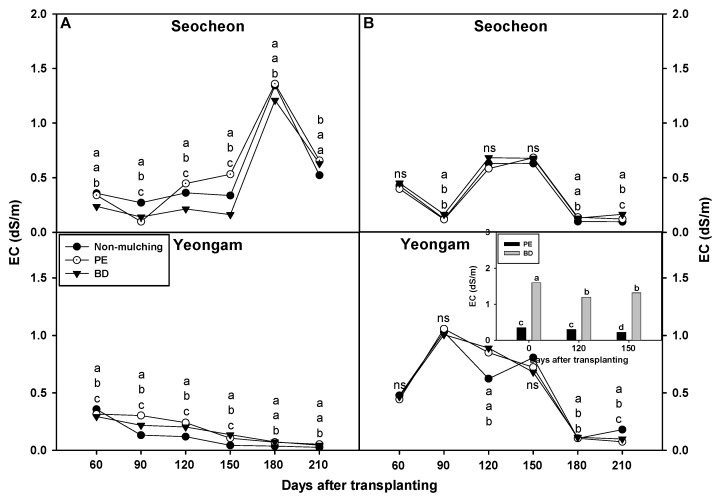
Effect of biodegradable (BD) and polyethylene (PE) films on soil EC in onion cultivation areas in 2023 (**A**) and 2024 (**B**). The values inside the figure represent the EC of the film solution at 120 and 150 days after transplanting in Yeongam, 2024. Within each box (representing an area), mean values with the same superscript letter(s) are not significantly different at the 5% level according to Duncan’s Multiple Range Test (DMRT). ns, non-significant.

**Figure 10 plants-14-02286-f010:**
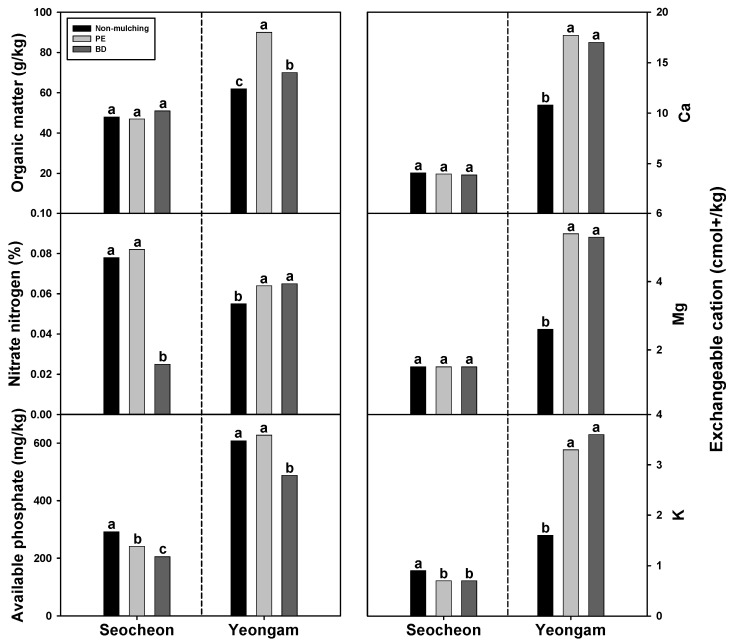
Effect of biodegradable (BD) and polyethylene (PE) films on soil nutrient levels 210 days after transplanting in onion cultivation areas in 2024. Within each box (representing an area), mean values with the same superscript letter(s) are not significantly different at the 5% level according to Duncan’s Multiple Range Test (DMRT).

**Figure 11 plants-14-02286-f011:**
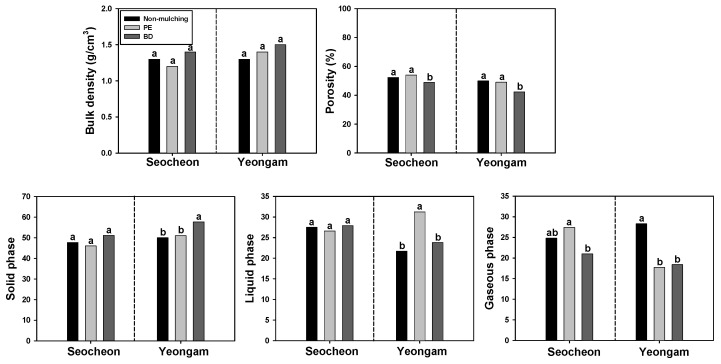
Effect of biodegradable (BD) and polyethylene (PE) films on soil bulk density, porosity, and the three-phase system 210 days after transplanting in onion cultivation areas in 2024. Within each box (representing an area), mean values with the same superscript letter(s) are not significantly different at the 5% level according to Duncan’s Multiple Range Test (DMRT).

**Figure 12 plants-14-02286-f012:**
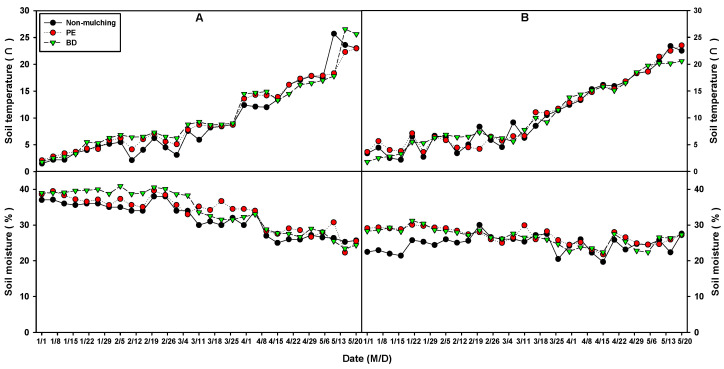
Soil temperature and soil moisture of biodegradable (BD) and polyethylene (PE) films in onion cultivation areas ((**A**), Seocheon; (**B**), Youngam) in 2024.

**Table 1 plants-14-02286-t001:** Nutrient levels in the experimental fields before the experiment in 2024.

Area	Organic Matter (%)	Nitrate Nitrogen (%)	Available Phosphate (mg/kg)	Exchangeable Cation (me/100 g)		
				Ca	Mg	K
Seocheon	5.75	0.02	429.83	7.85	1.84	1.08
Yeongam	7.53	0.03	214.43	4.88	1.97	0.69

**Table 2 plants-14-02286-t002:** Effect of biodegradable (BD) and polyethylene (PE) films on yield components and yield in onion cultivation areas in 2023 and 2024.

Year	Area	Film	Bolting	Bulb Height	Bulb Wide	Bulb Weight	Yield	Marketable Yield	Market Yield Rate
			(%)	(mm)	(mm)	(g)	(kg/ha)	(kg/ha)	(%)
2023	Seocheon	Non-mulching	0.0 ^a^	66.5 ^a^	72.0 ^a^	192.6 ^a^	44,268 ^a^	42,428 ^a^	95.8 ^a^
		PE	0.0 ^a^	71.9 ^a^	73.0 ^a^	205.4 ^a^	47,596 ^a^	46,029 ^a^	96.7 ^a^
		BD	0.0 ^a^	70.8 ^a^	73.2 ^a^	204.3 ^a^	48,526 ^a^	47,317 ^a^	97.5 ^a^
	Youngam	Non-mulching	3.3 ^a^	79.1 ^a^	84.1 ^b^	344.3 ^b^	53,040 ^a^	51,290 ^a^	96.7 ^a^
		PE	2.5 ^a^	80.0 ^a^	86.1 ^a^	350.0 ^a^	54,340 ^a^	52,982 ^a^	97.5 ^a^
		BD	1.7 ^a^	83.0 ^a^	90.0 ^a^	357.7 ^a^	55,380 ^a^	53,996 ^a^	97.5 ^a^
2024	Seocheon	Non-mulching	0.0 ^a^	62.0 ^b^	64.5 ^b^	130.1 ^b^	35,670 ^b^	27,820 ^a^	95.8 ^a^
		PE	0.0 ^a^	64.2 ^a^	67.9 ^a^	161.8 ^a^	49,670 ^a^	38,750 ^a^	96.7 ^a^
		BD	0.0 ^a^	66.5 ^a^	68.0 ^a^	163.0 ^a^	50,660 ^a^	39,510 ^a^	97.5 ^a^
	Youngam	Non-mulching	1.7 ^a^	65.3 ^a^	67.1 ^b^	162.1 ^b^	43,330 ^b^	42,430 ^a^	95.8 ^a^
		PE	0.8 ^a^	66.3 ^a^	69.6 ^a^	188.1 ^a^	51,010 ^a^	46,030 ^a^	96.7 ^a^
		BD	1.7 ^a^	68.3 ^a^	70.9 ^a^	186.9 ^a^	50,660 ^a^	45,950 ^a^	96.7 ^a^
Average		Non-mulching	1.3	68.2	71.9	207.3	44,077	40,992	96.0
		PE	0.8	70.6	74.2	226.3	50,654	45,948	96.9
		BD	0.9	72.2	75.5	228.0	51,307	46,693	97.3

Within each column (representing an area), mean values with the same superscript letter(s) are not significantly different at the 5% level according to Duncan’s Multiple Range Test (DMRT).

**Table 3 plants-14-02286-t003:** Effect of biodegradable (BD) and polyethylene (PE) films on weed occurrence rate (% of planting holes) in onion cultivation areas in 2024.

Area	Film	Weed Occurrence Rate (%)		
		Planting Hole	Degraded Film Area	Through Intact Film
Seocheon	Non-mulching	3.74 ^a^	0	0
	PE	1.31 ^b^	0	0
	BD	0.13 ^b^	0	0
Youngam	Non-mulching	5.13 ^a^	0	0
	PE	1.83 ^b^	0	0
	BD	2.92 ^b^	0	0

Within each column (representing an area), mean values with the same superscript letter(s) are not significantly different at the 5% level according to Duncan’s Multiple Range Test (DMRT).

**Table 4 plants-14-02286-t004:** Soil microbial population (CFU g^−1^) at 120 days after transplanting under biodegradable (BD) and polyethylene (PE) film treatment in onion cultivation areas (A, Seocheon; B, Youngam) in 2024.

Area	Film	Aerobic Bacteria	Bacillus Bacteria	Actinomycetes	Gram-Negative Bacteria	Filamentous Fungi
		(×10^5^)	(×10^3^)	(×10^4^)	(×10^4^)	(×10^3^)
Seocheon	Non-mulching	15.9 ^a^	34.5 ^b^	45.9 ^b^	26.3 ^a^	37.8 ^a^
	PE	15.0 ^a^	47.2 ^a^	53.3 ^a^	8.1 ^b^	37.2 ^a^
	BD	15.9 ^a^	25.7 ^b^	24.2 ^c^	14.0 ^b^	36.6 ^a^
Youngam	Non-mulching	37.4 ^a^	45.4 ^a^	38.1 ^b^	6.6 ^a^	10.6 ^b^
	PE	29.7 ^a^	57.1 ^a^	44.2 ^a^	6.3 ^a^	20.3 ^a^
	BD	11.3 ^b^	17.8 ^b^	36.8 ^b^	3.2 ^b^	8.1 ^c^

Within each column (representing an area), mean values with the same superscript letter(s) are not significantly different at the 5% level according to Duncan’s Multiple Range Test (DMRT).

## Data Availability

Data are contained within the article.

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
