# Peer review of "Assessment of the Agricultural Effectiveness of Biodegradable Mulch Film in Onion Cultivation"

_plants, 2025, doi:10.3390/plants14152286_

Round 1
Reviewer 1 Report (Previous Reviewer 2)
Comments and Suggestions for Authors
General comments:
When studies like this are made in different years and locations, the main objective is not comparing the detailed results among them. The relevant result is if the biodegradable film is comparable to PE film across a variety of environments. It is normal that the result is a little higher or lower according to the environment, but the average across years and locations is the only result that really matters. Probably, the few sporadic differences are just random effect that should not be highlighted. The lines 264-268 summarize the results and that was how the text should be. Presenting and discussing the results for each location and year is tiresome and pointless. You could discuss the difference between locations if you know the specific factor that caused the differences among environments, but that is not the case. This manuscript would be considerably more robust if the results were presented instead as the average across locations and years. When some significant difference was found, it could be mentioned, particularly if there is a reason or some explanation why it was observed.
The presentation of the results is very difficult to read because the study has a large number of comparisons (2 locations x 2 years x 5 sampling times). In cases like this, the reader gets very confused if the authors comments on many specific comparisons. Like for the results on plant height, it would be friendly to present the readers with an overall results and then you can just make some comments on the exceptions (e.g., “Overall, the plant height was not influenced in Seocheon, while in Yeongan, the treatment without mulching resulted in shorter plants. In Seocheon, there were significant differences in plant height only in one measurement in 2023 (at 150 DAP) and in 2024 (at 180 DAP), but the differences were no longer detected in the following measurements.”). Please consider making this simplification not only in this paragraph, but all the section of results and discussion should be revised doing your best for making the text easier to read.
Figures 1, 9 and 10: It is interesting to display the letter of means separation test in the plotting area; however, it is challenging to the reader to guess to which data point each letter refers. It may seem intuitive to the authors that prepared the graph, but it is not informed anywhere how the letters are organized. It is like a puzzle, and I took a long time to find the logic. Please find a way to present the means separation test in a comprehensible way. For instance, when there is not difference between treatments, you should just display the traditional “ns”; when there is difference, you could match the vertical position of the data points with the vertical sorting of the letters. In other words, the letter placed on the bottom refers to the data point that has the smaller value. It is more intuitive that the letter follows the same sequence that the data points are plotted.
Table 2: It would be great to include a line with the average across years and locations for each treatment.
Line 287: It is informed that residues of film were collected from the NM plots. As that is impossible, please correct the sentence.
Figure 2: It is not usual to adopt the unit as “g/ 3 plants”. You should divide the result by 3 and present it as “g/plant”.
Figure 4: The data point of 2023-BD-210 DAT was placed precisely in the break of the vertical axis. Then, the value is anything from 1 to 4. Please correct this imprecision. Maybe the brake could be placed between 1.5 and 4.5. Even better would be presenting the graph with the vertical axis varying from 0 to 1.5 instead of 5.
Comments on the Quality of English LanguageLine 38: Delete the word “globally” or rewrite the sentence to fit the word. It is missing a word after “common”: …has become a common agricultural practice in numerous countries,…”
Line 48: Check the spelling of “, biodigradable"
Line 128: Check the spelling of “fregmants”
Line 175: “…10 g of soil were placed…
Line 241: NM equals to “no treatment”. Therefore, it is awkward when you write “treated with no treatment”. It would be better if you could say “the crops without mulch…” [please, consider revising the whole manuscript for that this suggestion, not only this phrase.]
Line 259: This is redundant. Yield is implicitly a production per area. It is enough to write only yield (delete “per hectare”). Or replace it with “onion yield”.
Lines 273-274: Please rephrase this sentence. It is fragmented and difficult to understand. (e.g.: Weeds did not emerge neither on the areas where the film collapsed nor in the areas covered with the film). “Film itself” is not an adequate expression. Please replace it with a better description.
Line 288: replace “seeded” with “sown”.
Lines 456-458. Check the font size and type. It seems to be different.
Author Response
General comments:
When studies like this are made in different years and locations, the main objective is not comparing the detailed results among them. The relevant result is if the biodegradable film is comparable to PE film across a variety of environments. It is normal that the result is a little higher or lower according to the environment, but the average across years and locations is the only result that really matters. Probably, the few sporadic differences are just random effect that should not be highlighted. The lines 264-268 summarize the results and that was how the text should be. Presenting and discussing the results for each location and year is tiresome and pointless. You could discuss the difference between locations if you know the specific factor that caused the differences among environments, but that is not the case. This manuscript would be considerably more robust if the results were presented instead as the average across locations and years. When some significant difference was found, it could be mentioned, particularly if there is a reason or some explanation why it was observed.
The presentation of the results is very difficult to read because the study has a large number of comparisons (2 locations x 2 years x 5 sampling times). In cases like this, the reader gets very confused if the authors comments on many specific comparisons. Like for the results on plant height, it would be friendly to present the readers with an overall results and then you can just make some comments on the exceptions (e.g., “Overall, the plant height was not influenced in Seocheon, while in Yeongan, the treatment without mulching resulted in shorter plants. In Seocheon, there were significant differences in plant height only in one measurement in 2023 (at 150 DAP) and in 2024 (at 180 DAP), but the differences were no longer detected in the following measurements.”). Please consider making this simplification not only in this paragraph, but all the section of results and discussion should be revised doing your best for making the text easier to read.
Response: We thank the reviewer for the helpful suggestion to simplify the presentation of the Results and Discussion sections. As recommended, we revised the text to provide an overall synthesis of the results first, followed by brief mentions of key exceptions where statistically significant differences were observed. This approach was applied to the plant height section as well as to other results including yield, microbial populations, and film degradation. To improve readability, we removed repetitive statistical comparisons and instead emphasized consistent trends across regions, years, and treatments. Specific details and exceptions were retained only where they added meaningful context. We believe these changes have significantly improved the clarity and accessibility of the manuscript. The revised text is shown in red in the updated manuscript for your review. We appreciate your constructive feedback, which helped us enhance the overall quality of the presentation.
Figures 1, 9 and 10: It is interesting to display the letter of means separation test in the plotting area; however, it is challenging to the reader to guess to which data point each letter refers. It may seem intuitive to the authors that prepared the graph, but it is not informed anywhere how the letters are organized. It is like a puzzle, and I took a long time to find the logic. Please find a way to present the means separation test in a comprehensible way. For instance, when there is not difference between treatments, you should just display the traditional “ns”; when there is difference, you could match the vertical position of the data points with the vertical sorting of the letters. In other words, the letter placed on the bottom refers to the data point that has the smaller value. It is more intuitive that the letter follows the same sequence that the data points are plotted.
Response: We thank the reviewer for the helpful suggestion regarding the clarity of the mean separation lettering. As recommended, we revised the figures to align letters with the vertical order of data points and added “ns” where no significant differences were observed. We believe this adjustment makes the figures more intuitive and easier to interpret.
Table 2: It would be great to include a line with the average across years and locations for each treatment.
Response: Included the average across years and locations for each treatment.
Line 287: It is informed that residues of film were collected from the NM plots. As that is impossible, please correct the sentence.
Response: Corrected.
Figure 2: It is not usual to adopt the unit as “g/ 3 plants”. You should divide the result by 3 and present it as “g/plant”.
Response: Corrected to “g/plant”.
Figure 4: The data point of 2023-BD-210 DAT was placed precisely in the break of the vertical axis. Then, the value is anything from 1 to 4. Please correct this imprecision. Maybe the brake could be placed between 1.5 and 4.5. Even better would be presenting the graph with the vertical axis varying from 0 to 1.5 instead of 5.
Response: Corrected.
Comments on the Quality of English Language
Line 38: Delete the word “globally” or rewrite the sentence to fit the word. It is missing a word after “common”: …has become a common agricultural practice in numerous countries”
Response: Corrected.
Line 48: Check the spelling of “, biodigradable"
Response: Corrected.
Line 128: Check the spelling of “fregmants”
Response: Corrected.
Line 175: “…10 g of soil were placed…
Response: Corrected.
Line 241: NM equals to “no treatment”. Therefore, it is awkward when you write “treated with no treatment”. It would be better if you could say “the crops without mulch…” [please, consider revising the whole manuscript for that this suggestion, not only this phrase.]
Response: Corrected.
Line 259: This is redundant. Yield is implicitly a production per area. It is enough to write only yield (delete “per hectare”). Or replace it with “onion yield”.
Response: Corrected.
Lines 273-274: Please rephrase this sentence. It is fragmented and difficult to understand. (e.g.: Weeds did not emerge neither on the areas where the film collapsed nor in the areas covered with the film). “Film itself” is not an adequate expression. Please replace it with a better description.
Response: Corrected.
Line 288: replace “seeded” with “sown”.
Response: Corrected.
Lines 456-458. Check the font size and type. It seems to be different.
Response: Corrected.
Reviewer 2 Report (New Reviewer)
Comments and Suggestions for Authors
Brief Summary: This is a comprehensive study of the effects of a biodegradable mulch film on onion growth, yield, and soil environmental parameters as compared to a polyethylene mulch film or no mulching. The results indicate no overall differences between these mulching strategies, with the exception of reduced microbial populations beneath the biodegradable mulch film. This is valuable work and adds to the body of literature supporting use of more sustainable products during production of onion and other agricultural crops.
General Concept Comments: The manuscript was mostly clearly presented, except for the notable lack of detail regarding experimental design and statistical methods (lines 216-218). It was a completely randomized experimental design, and apparently ANOVA and t-tests were used to perform means separations. However, it is unclear how the plots were arranged, how many replicates there were for the different parameters, and how many plants constituted an experimental unit. Test plots were described as a “minimum area of 50 m x 20 m per treatment.” How many test plots were there per treatment? And how many onion plants were in each plot? In lines 117-118, it says, “Plant height and yield components were measured with 10 plants for each treatment with three replications.” Does that mean there were only 10 plants in each plot and three plots of each mulch type? Or were only 10 plants from each treatment sampled from larger plots of plants? (If so, how were those plants selected so that they would be representative of that treatment?) Were the rows in each plot mulched differently, or was the entire plot in one type of mulch? A diagram might be helpful. It would also be helpful to have “n=” in the table headings or figure captions to indicate the sample size.
Specific Comments:
Lines 57-58: PHA and PHB should be defined.
Line 80: What is MgOB?
Line 116: Yield is based on a minimum bulb size of 3 cm diameter. What kind of onions were these? The mature bulb size seems small (<3cm). I could find no information on the varieties listed.
Lines 120-121: 'The number of weeds were assessed in each of 1,000 planting holes.' Where is this data? Table 3 reports weed occurrence as %, not number of weeds.
Line 149: Do you mean weight of the film measured at each ‘days after transplanting time point’ rather than ‘irradiation time point’?
Line 165: What did the visual assessment consist of? Is this the score rating?
Lines 216-218: This needs more detail.
Line 274: What are ‘film collapse sites’? Terminology for this differs throughout the manuscript.
Line 276: What is meant by ‘4-5% of weeds emerged’? Percent of what?
Figure 2 caption is confusing. What is ‘post-crop (soybean) soil collected from onion cultivation areas’? Do you mean ‘post-crop (soybean) planted in soil collected from onion cultivation areas’?
Table 3: Confusing. It indicates weed occurrence rate as a percent. Percent of what? Percent of total area? What is the ‘film collapse site’? Is it separate from the onion plots? Weed occurrence rate in the ‘film itself’ – does this mean above the film? Below the film? Through the film?
Line 355: Do you mean ‘score level 1’?
Figure 4: ‘visual collapse levels’ – this is different from the terminology used in the Methods. Does it refer to the visual score levels?
Figure 5: Necessary? The photos are hard to see and don’t convey any meaningful data. Caption: Film collapse level – this terminology is different from Figure 4. Is this a visual representation of what is presented in Figure 4?
Line 587: By ‘floor area density’ do you mean ‘soil bulk density’?
Author Response
Reviewer 2
Brief Summary: This is a comprehensive study of the effects of a biodegradable mulch film on onion growth, yield, and soil environmental parameters as compared to a polyethylene mulch film or no mulching. The results indicate no overall differences between these mulching strategies, with the exception of reduced microbial populations beneath the biodegradable mulch film. This is valuable work and adds to the body of literature supporting use of more sustainable products during production of onion and other agricultural crops.
General Concept Comments: The manuscript was mostly clearly presented, except for the notable lack of detail regarding experimental design and statistical methods (lines 216-218). It was a completely randomized experimental design, and apparently ANOVA and t-tests were used to perform means separations. However, it is unclear how the plots were arranged, how many replicates there were for the different parameters, and how many plants constituted an experimental unit. Test plots were described as a “minimum area of 50 m x 20 m per treatment.” How many test plots were there per treatment? And how many onion plants were in each plot? In lines 117-118, it says, “Plant height and yield components were measured with 10 plants for each treatment with three replications.” Does that mean there were only 10 plants in each plot and three plots of each mulch type? Or were only 10 plants from each treatment sampled from larger plots of plants? (If so, how were those plants selected so that they would be representative of that treatment?) Were the rows in each plot mulched differently, or was the entire plot in one type of mulch? A diagram might be helpful. It would also be helpful to have “n=” in the table headings or figure captions to indicate the sample size.
Response: We appreciate the reviewer’s thoughtful feedback regarding the need for more detailed information on the experimental design and statistical analysis. In response, we have revised the Materials and Methods section to clearly describe the plot layout, replication structure, and sampling procedures. Specifically, we clarified that the experiment followed a completely randomized design with three replications per treatment. For yield measurements, the experimental unit consisted of plants grown within a 3.3 m² area, selected from within each replication plot. Three such replications were established per treatment. We also specified that plant height and yield components were measured using 10 randomly selected plants per replicate, sampled from larger plots to ensure representativeness. Each plot was uniformly treated with a single mulch type (BD, PE, or NM), and we clarified that no split-row treatments were applied. We believe these revisions improve the transparency and reproducibility of the study and thank the reviewer for this valuable recommendation.
Specific Comments:
Lines 57-58: PHA and PHB should be defined.
Response: Defined in line 53-54
Line 80: What is MgOB?
Response: Definition added.
Line 116: Yield is based on a minimum bulb size of 3 cm diameter. What kind of onions were these? The mature bulb size seems small (<3cm). I could find no information on the varieties listed.
Response: Thank you for your valuable observation. As listed in the manuscript, the onion varieties used in this study were Oreo" and "Katamaru. These are commonly cultivated commercial varieties in the study region, and their mature bulb size typically exceeds 3 cm in diameter. In our study, marketable yield was calculated based on bulbs with a diameter of 3 cm or larger, as this is the standard commercial threshold in the local market. Importantly, the proportion of bulbs below this threshold was minimal—marketable yield accounted for over 96%, indicating that bulbs smaller than 3 cm made up only approximately 4% of the total harvest. Thus, the yield data presented in the manuscript accurately reflect the performance of mature, marketable onion bulbs.
Lines 120-121: 'The number of weeds were assessed in each of 1,000 planting holes.' Where is this data? Table 3 reports weed occurrence as %, not number of weeds.
Response: Corrected.
Line 149: Do you mean weight of the film measured at each ‘days after transplanting time point’ rather than ‘irradiation time point’?
Response: Corrected.
Line 165: What did the visual assessment consist of? Is this the score rating?
Response: Addressed.
Lines 216-218: This needs more detail.
Response: Addressed in revised Materials and Methods.
Line 274: What are ‘film collapse sites’? Terminology for this differs throughout the manuscript.
Response: Terminology standardized throughout manuscript.
Line 276: What is meant by ‘4-5% of weeds emerged’? Percent of what?
Response: Changed.
Figure 2 caption is confusing. What is ‘post-crop (soybean) soil collected from onion cultivation areas’? Do you mean ‘post-crop (soybean) planted in soil collected from onion cultivation areas’?
Response: Changed.
Table 3: Confusing. It indicates weed occurrence rate as a percent. Percent of what? Percent of total area? What is the ‘film collapse site’? Is it separate from the onion plots? Weed occurrence rate in the ‘film itself’ – does this mean above the film? Below the film? Through the film?
Response: Corrected.
Line 355: Do you mean ‘score level 1’?
Response: Corrected.
Figure 4: ‘visual collapse levels’ – this is different from the terminology used in the Methods. Does it refer to the visual score levels?
Response: Corrected.
Figure 5: Necessary? The photos are hard to see and don’t convey any meaningful data. Caption: Film collapse level – this terminology is different from Figure 4. Is this a visual representation of what is presented in Figure 4?
Response: Thank you for your observation. We agree that Figure 5 did not clearly convey additional data beyond what is already presented in Figure 4. Therefore, we have removed Figure 5 to streamline the presentation and avoid redundancy.
Line 587: By ‘floor area density’ do you mean ‘soil bulk density’?
Response: Corrected.
Reviewer 3 Report (New Reviewer)
Comments and Suggestions for Authors
Dear Authors,
This study aims to address a relevant topic in sustainable agriculture, employing a strong multi-site, multi-year approach. However, the manuscript would benefit from clearer interpretation of microbial impacts, regional variability, and non-significant results. Adding a balanced discussion of limitations and ecological implications would enhance its value. Good luck with the revisions.

Author Response
Reviewer 3
Dear Authors,
This study aims to address a relevant topic in sustainable agriculture, employing a strong multi-site, multi-year approach. However, the manuscript would benefit from clearer interpretation of microbial impacts, regional variability, and non-significant results. Adding a balanced discussion of limitations and ecological implications would enhance its value. Good luck with the revisions.
- Abstract
The abstract includes useful information, but could be more concise and impactful. The key findings are somewhat buried, and the structure would benefit from focusing more sharply on outcomes and their broader significance. I suggest trimming repetitive background lines and bringing forward the main results, especially yield performance and microbial changes. Quantitative data (e.g., % yield increase, degradation rates) should be added for clarity. Additionally, the environmental relevance of using biodegradable films should be summarised in the final sentence for stronger closure.
Response: Thank you for your constructive feedback. We revised the abstract to make it more concise and impactful by trimming background information and emphasizing the key quantitative findings (e.g., yield, degradation). The final sentence now highlights the environmental significance of using biodegradable films in sustainable agriculture.
- Introduction
The background on mulching and biodegradable plastics is well laid out. However, the introduction does not clearly explain why onion was chosen as a model crop or how this study adds to the body of knowledge. Please consider refining the justification for selecting onion cultivation as the test system. It would also help to identify what gaps this study addresses—e.g., long-term soil health impacts or microbial changes related to BD film use, which are currently underexplored. Ending this section with a more precise research aim would guide the reader better.
Response: We thank the reviewer for this suggestion. We revised the Introduction to justify the selection of onion due to its sensitivity to mulch effects, economic relevance, and the role of mulch in managing soil moisture and weeds. We also outlined the knowledge gaps addressed in this study—particularly the underexplored long-term effects on soil microbial communities. The final paragraph now ends with a more precise research objective.
- Materials and Methods
The methodology is comprehensive, covering field design, treatments, and measurements. That said, some parts go into unnecessary technical detail (e.g., brand names, hole sizes, etc.) which could be condensed or placed in supplementary materials.
Response: Thank you for your feedback. While we retained key technical specifications for reproducibility, we reviewed the section to streamline content and ensured that only essential manufacturer details were kept. If needed, we are open to relocating technical details to supplementary material.
- Results and Discussion
4.1 Growth and Yield
The data clearly demonstrate that both BD and PE films supported onion growth better than non-mulched controls. However, several results labeled as "not significantly different" are followed by conclusions that imply treatments were equally effective. I urge caution in this interpretation. Rather than implying treatments are equivalent, it’s more appropriate to state that no significant differences were detected. This is particularly important when discussing yield components, as the trend might be agronomically relevant even if not statistically significant.
Response: Thank you for this important observation. We have revised the language to avoid implying equivalence based on non-significant results. Instead, we now clearly state that “no significant differences were detected.” We also acknowledged agronomically relevant trends where appropriate. These edits have been applied throughout Section 3.1.
4.2 Weed Suppression
The weed data support the practical benefit of mulching, though the results between BD and PE are similar. It might help to briefly note whether the weed control differences—though small—have management implications. Also, mentioning if any dominant weed species were resistant to BD films would be of interest.
Response: We expanded the discussion to explain that the slightly higher weed emergence under BD (0–3%) compared to PE (1–2%) is not expected to impact management practices. Additionally, we clarified that no dominant weed species were resistant to BD film. These revisions enhance both clarity and practical interpretation.
4.3 Soybean Post-Crop Results
One of the more novel aspects of this work is the post-crop performance assessment. However, the results vary considerably between Seocheon and Yeongam, and the discussion doesn’t explore these differences. These contrasting outcomes deserve more explanation. Could they be due to initial soil fertility, BD degradation byproducts, or microbial community shifts? A few lines addressing these possibilities would significantly improve this section.
Response: We appreciate the reviewer’s suggestion. We expanded this section to explore factors contributing to regional differences, including initial soil fertility, microbial activity, and BD film degradation rates. We noted that no adverse effects of BD byproducts were observed, but future research is needed to explore these interactions further.
4.4 Degradation Analysis
The study nicely tracks BD film breakdown across time and soil conditions. Light transmittance, visual scoring, and weight loss methods all reinforce the conclusion that BD films degrade substantially during the growing season. One area to expand is whether early degradation affects the mulching benefits (e.g., weed control or moisture conservation). It may also be useful to reflect on how degradation rates might differ under warmer summer crops, as degradation seems slower in cooler conditions here.
Response: In response, we added discussion on how early-stage degradation could potentially reduce the duration of mulching benefits like weed control or moisture retention—especially in warmer crops. We emphasized the importance of matching film degradation rates to crop cycles and environmental conditions.
4.5 Soil Properties and Microbiology
This section presents important findings, especially the reduced microbial populations under BD films, specifically Bacillus and actinomycetes. However, this aspect is under-discussed despite its significance. Please consider expanding on what reduced microbial counts might imply. Could this affect nutrient cycling, disease suppression, or long-term soil health? Even if short-term impacts seem minimal, the ecological implications over repeated seasons could be substantial. Bringing in relevant literature here would strengthen the discussion.
Response: Thank you for highlighting this. We have greatly expanded the microbial discussion to address potential implications for nutrient cycling, disease suppression, and long-term soil health. We also incorporated relevant references (e.g., soil microbial shifts from BD use) to support these points and emphasize the importance of further long-term research.
- Conclusion
The conclusion captures the overall findings, but it feels more like a summary than a synthesis. There’s also little critical reflection. Please consider adding a few lines about study limitations. For example, only two locations were tested, the study spanned only two seasons, and the degradation scoring was subjective. Also, suggesting directions for further research such as long-term studies on BD residues, microbial dynamics, or cost-benefit assessments would add real value.
Response: We revised the conclusion to include critical reflection on study limitations, such as the two-season, two-location scope and subjectivity in degradation scoring. We also added suggestions for future research, including BD film residue persistence, microbial impacts over time, and cost-benefit assessments. These changes better synthesize the findings and emphasize broader relevance.
Round 2
Reviewer 3 Report (New Reviewer)
Comments and Suggestions for Authors
Good Luck.
Author Response
We thank the reviewer for their time and positive evaluation of our manuscript. We appreciate your support.
This manuscript is a resubmission of an earlier submission. The following is a list of the peer review reports and author responses from that submission.
Round 1
Reviewer 1 Report
Comments and Suggestions for Authors
The manuscript "Assessment of the agricultural effectiveness of biodegradable mulch film in onion cultivation" addresses a important topic concerning the possibility of using biodegradable films in onion cultivation. The results of these studies confirm that synthetic films can be replaced by biodegradable films.
In its current version, it is not suitable for publication in Plants journal. The manuscript should be corrected and supplemented. The main objection is that some of the presented results come from one-year studies. In the article does not present the weather conditions in 2023 and 2024.
Anticipated the fact that weather conditions (temperature, humidity, precipitation) have a very strong influence on the condition of soil and plants, as well as on the rate of decomposition of biodegradable film, such studies should be conducted over a period of 2-3 years.
The authors provide conclusions regarding the condition of soil (chemical composition, organic matter, microorganisms) after one-year studies. They do not include in their studies such important factors as soil type and weather conditions, which contribute to a large extent to plant growth, as well as the condition of soil and litter, especially biodegradable film.
Weed infestation was assessed in only one year? The condition of weeds may change under the effect of, for example, weather conditions, and therefore changes in the years of the study.
Line 62. Full, scientific name of the species i.e. Codonopsis lanceolata (Siebold & Zucc.) Benth. & Hook.f. ex Trautv.
Line 83. What do the distances given here refer to?
How were the onion seedlings planted? manually or mechanically (what kind of machine was used?)? How were the holes made in the mulches?
- No soil characteristics were given (soil type, chemical composition, pH, EC) in both study locations.
The authors did not describe the weather conditions in the years of the study. This is an important factor (temperature, precipitation) contributing to the rate of decomposition of the biodegradable film.
The information on onion fertilization should be explained and supplemented
- Why were chemical analyses of the soil only performed in 2024?
- If in 2023, before the experiment was established, no analyses of the chemical composition of the soil were performed, on what basis the dose of fertilizer for onions determined?
- What is the assumed optimal content of N, P, K, Ca in the soil for onions?
- Was the same dose of fertilizer applied before onion cultivation in both places of research, despite the different chemical composition of the soil?
The units should be recorded in accordance with generally accepted principles. Standardize the units.
Line 117. How were such small pieces of films was obtained?
Due to the lack of differences or minor differences in the characteristics of the plants, the reviewer believes that these research results can be presented as averages from two years. Especially since the authors did not present the weather conditions prevailing in subsequent years in both places.
Lines 444 – 447 – the sentence does not concern the topic discussed in this chapter.
Chapter 3.3. is devoted to the influence of fims mulches on soil pH and EC. The authors describe in great detail the changes in soil pH and EC under different mulches. They state that PE and BD had no influence on these parameters. Where does this conclusion come from? – in the beginning of experiment of the soil for the two research sites was not given. There is no data on soil pH and EC in 2023 and 2024 before the experiment was established.
There was no discussion in this chapter.
Line 489 – 493 Does the paragraph concern the EC of PE and BD films or the EC of soil under PE or BD film mulch? Please correct.
The discussion of research results was conducted poorly. Very few publications by other authors were used in the discussion. Some results, e.g. regarding tensile strength, were not compared with other publications.
Reviewer 2 Report
Comments and Suggestions for Authors
Line 40: to be precise, mulching is not for regulating temperature, but always for warming soil temperature.
Line 43, 48: In this case, do not use BD as abbreviation for the word biodegradable. I think you should not use abbreviations at all, but at least restrict its use to the “treatment of biodegradable mulch”.
Line 117: Just add the detail if soybean was sown immediately after mixing the mulch fragments or it occurred after some weeks. It is also important to inform the area of the pot because the plant density on the pot influences the plant height. Also clarify if shoot height and weight were measured 14 days after sowing or 14 days after measuring germination.
Line 131: “…applying a scale of 0 to 5. The score 0 indicates…” replace the word “step” with “score”.
Lines 140-141: Please rewrite this sentence. I could not understand what was measured and when it was measured; Delete the word “interval” and write “at 150 days after transplantation”.
Line 152: Replace the word experiment with measurements (e.g., “…with measurements made in triplicate.”)
Line 157: “Effects… on film soil properties
Line 205: “2.5 Experimental design and statistical procedures”
Section 3.2: Please check if what is being presented is the light transmittance or absorption. It is being presented that the film light transmittance is around 1%, but it seems more plausible that the film transmits 99% and absorbs 1%. It should be the opposite. If that is correct, please revise the text and figures.
Figure 7: The units on the vertical axis (tensile strength and elongation) do not need to be presented with four decimal points (it could be 1% instead of 1.000%). By the way, replace commas with points.
Figure 11 is difficult to read. Please move the nutrient to inside the plot area instead presenting them outside. Looking to the figure I took a long time to understand that the columns on the left are for organic matter and in the right are for calcium. To avoid that problem, the graphs for each nutrient should be separated (it cannot be a continuous line like in the current version). They can be arranged in the same figure, but they need to be clearly separated. The combination of dashed / continuous lines does not work for that purpose.
Comments on the Quality of English LanguageLine 38: suggestion for choice of word: “…has become a common agricultural practices…” (staple usually means basic food).
Line 105: delete the word “intervals”
Please revise (and avoid) the use of the prefixes “post- pre-“, and use the words “after or before” instead. It is not usual the use as “post-onion”, “pre-mulching”, “post-mulching”, “post-crop” etc.
Please revise the text for the situations in which “NM” was written as an applied treatment. I understand that it can be written like that, but this style is confusing for the reader. It would be clear if you write “without mulch”. For example (line 280), instead of writing “areas that used NM”, it reads quite better “areas without mulch”. To write even better, the final of the sentence could be “compared with the soil covered with PE or BD films”.
Reviewer 3 Report
Comments and Suggestions for Authors
The manuscript presents a study evaluating the use of biodegradable mulching films (BD) in onion cultivation, comparing them with conventional polyethylene (PE) films and non-mulched controls.
The research encompasses a comprehensive analysis of several key parameters, including plant growth and yield, soil environmental conditions, weed suppression efficacy, film degradation, and the impact on soil microbial communities. The results indicate that the BD films have strong potential as a sustainable alternative to PE films, demonstrating comparable agronomic effectiveness while offering the advantage of biodegradability. Moreover, the authors rightly emphasize the need for further research into the durability, decomposition rates, and economic feasibility of BD films for large-scale adoption.
A notable strength of the manuscript is the well-structured and up-to-date literature review, which provides a solid theoretical foundation for the study—31 out of 50 cited references are from the past decade. The methodology is clearly described and appropriate for the research goals. Particularly commendable are the figures, which are clear, well-designed, and enhance the reader’s understanding of the presented data.
In my opinion, the manuscript meets the standards required for publication in a scientific journal and makes a meaningful contribution to the field of sustainable agricultural practices. I recommend the manuscript for publication in its current form / subject to minor editorial revisions.